# MSC Based Therapies to Prevent or Treat BPD—A Narrative Review on Advances and Ongoing Challenges

**DOI:** 10.3390/ijms22031138

**Published:** 2021-01-24

**Authors:** Maurizio J. Goetz, Sarah Kremer, Judith Behnke, Birte Staude, Tayyab Shahzad, Lena Holzfurtner, Cho-Ming Chao, Rory E. Morty, Saverio Bellusci, Harald Ehrhardt

**Affiliations:** 1Department of General Pediatrics and Neonatology, Universities of Giessen and Marburg Lung Center (UGMLC), Member of the German Center for Lung Research (DZL), Justus-Liebig-University, Feulgenstrasse 12, 35392 Giessen, Germany; maurizio.j.goetz@med.uni-giessen.de (M.J.G.); sarah.kremer@paediat.med.uni-giessen.de (S.K.); judith.behnke@paediat.med.uni-giessen.de (J.B.); birte.staude@med.uni-giessen.de (B.S.); tayyab.shahzad@paediat.med.uni-giessen.de (T.S.); lena.holzfurtner@med.uni-giessen.de (L.H.); cho-ming.chao@paediat.med.uni-giessen.de (C.-M.C.); 2Department of Internal Medicine II, Universities of Giessen and Marburg Lung Center (UGMLC), Cardiopulmonary Institute (CPI), Member of the German Center for Lung Research (DZL), Justus-Liebig-University, Aulweg 130, 35392 Giessen, Germany; saverio.bellusci@innere.med.uni-giessen.de; 3Department of Lung Development and Remodeling, Max Planck Institute for Heart and Lung Research, Member of the German Center for Lung Research (DZL), Ludwigstrasse 43, 61231 Bad Nauheim, Germany; rory.morty@mpi-bn.mpg.de

**Keywords:** chronic lung disease, bronchopulmonary dysplasia, preterm, mesenchymal stem cells, lung development, inflammation, lung injury, lung repair, extracellular vesicles

## Abstract

Bronchopulmonary dysplasia (BPD) remains one of the most devastating consequences of preterm birth resulting in life-long restrictions in lung function. Distorted lung development is caused by its inflammatory response which is mainly provoked by mechanical ventilation, oxygen toxicity and bacterial infections. Dysfunction of resident lung mesenchymal stem cells (MSC) represents one key hallmark that drives BPD pathology. Despite all progress in the understanding of pathomechanisms, therapeutics to prevent or treat BPD are to date restricted to a few drugs. The limited therapeutic efficacy of established drugs can be explained by the fact that they fail to concurrently tackle the broad spectrum of disease driving mechanisms and by the huge overlap between distorted signal pathways of lung development and inflammation. The great enthusiasm about MSC based therapies as novel therapeutic for BPD arises from the capacity to inhibit inflammation while simultaneously promoting lung development and repair. Preclinical studies, mainly performed in rodents, raise hopes that there will be finally a broadly acting, efficient therapy at hand to prevent or treat BPD. Our narrative review gives a comprehensive overview on preclinical achievements, results from first early phase clinical studies and challenges to a successful translation into the clinical setting.

## 1. Introduction

Taking embryological lung development into mind, preterm infants born ≤32 weeks of gestation have their lungs in the late canalicular and saccular stages of lung development. After birth, most of these infants depend on respiratory support and oxygen supply due to lung immaturity and surfactant deficiency. They are exposed to pre- and postnatal infections, antibiotic therapy and a potentially harmful microbial environment in the NICU. Physiologic nutritional supply via the umbilical cord is interrupted and metabolic processes can be deranged by immaturity and stress. Furthermore, genetic predispositions, intrauterine growth restriction, smoke exposure and necessary clinical measures including fluid supply to stabilize the cardiovascular system need to be enumerated as potentially harmful. All these factors are highly recognized to constitute risk factors for restrictions of physiologic lung development in this critical period resulting in lifelong persisting limitations in lung function called bronchopulmonary dysplasia (BPD) [1,2,3,4,5]. From a histopathologic perspective, BPD is characterized by the distortion of epithelial and vascular development and of the extracellular matrix composition. Most studies evaluated the impact of mechanical ventilation, oxygen toxicity and bacterial infections on disease development. They identified the inflammatory response in the immature lung initiated before birth or shortly after birth to cause an imbalance of growth factors and anti-inflammatory cytokines on the one side and pro-inflammatory activation on the other [1]. One key feature of disease pathogenesis represents the dysfunction and rarefication of mesenchymal stem cells (MSC) in the lung that has been reviewed in detail by us recently [6]. Briefly, in the physiologic situation, MSC are located mainly at the tips of the secondary septa and orchestrate the complex interplay between epithelium and endothelium. The inflammatory response leads to fundamental changes in MSC phenotype and function and MSC rarefication. That has been described in rodent models in detail and was confirmed in lungs from deceased infants and in studies on tracheal aspirates of preterm infants [7,8,9]. These congruent data fundaments the therapeutic potential of strategies to reprogram resident lung MSC or to substitute them by exogenous transfer into the lung. While therapeutic reprogramming is just an upcoming issue faced with many obstacles and uncertainties despite the advantage of the full therapeutic potential due to the lung specific phenotype, the multitude of successful studies on MSC application to the injured lung in preclinical models raises hopes that there will be a more efficient strategy at hand in the near future than the currently available therapeutics used in preterm infants [7,10]. These therapies with proven efficacy in meta-analyses and systematic reviews comprise exogenous surfactant application, vitamin A, caffeine, azithromycin and postnatal corticosteroids summarized in Table 1. Despite the initial enthusiasm about for instance exogenous surfactant and the immediate tremendous benefits on gas exchange, they all have in common an only modest impact on the outcome of BPD [11,12,13,14,15,16,17,18,19]. Ureaplasma and mycoplasma species are hold responsible to aggravate inflammation and lung injury but their eradication by macrolide antibiotics remains a clinical challenge and inflammation is not only fostered by these pathogens but by the life-saving therapies of mechanical ventilation and oxygen supply. Despite, azithromycin but not erythromycin reduced the risk of BPD that might be ascribed to its overall anti-inflammatory properties [15,20]. Corticosteroids as the last class of drugs to be mentioned here are well-known for their impressive anti-inflammatory action in animal studies that accounts for one of the key mechanisms to preserve the immature lung when given prenatally. Identical effects are evident when given postnatally. However, their use needs to be restricted to most severely affected infants since they are known to impair the long-term outcome: Negatively affected psychomotor development is the best studied parameter while potential risks for somatic growth, endocrine homeostasis, cardiovascular function and metabolic diseases later in life still await a better database [20,21,22,23,24].

These considerations argue for a thorough wrap up of the existing data on MSC based therapies to prevent BPD. BPD originates from intrauterine and early postnatal pathologies, therefore we put special considerations to the therapeutic potential to treat developing BPD as pursued in most preclinical and clinical studies.

## 2. What Are the Proven Benefits of MSC Application within the Injury-Repair-Regeneration Cascade?

MSC have been studied in nearly all relevant diseases in preclinical animal models. MSC have in common that they mainly act via the release of anti-inflammatory cytokines and growth factors that have the potential to at least partially revert the inflammatory response in the damaged and inflamed organ which constitutes the initial phase of an evolving disease. Therefore, the biggest effects were detected in animal models when applied simultaneously or shortly after injury but due to their growth and repair promoting properties, therapeutic benefits can be detected even for the retarded application depending on the disease entity and disease status [10].

Both the innate and adaptive cellular immune response get reverted towards the physiologic status and further immune cell attraction is prohibited. Thereby, MSC have the potential to redirect any leukocyte population towards beneficial phenotypes, including the shift from M1 to M2 macrophages, the propagation of Treg T cells and the inhibition of Th17 response, neutrophil extracellular trap formation and modulation of antigen presenting cells like B cells and dendritic cells. The attenuation of inflammation and protease action is completed by the release of further beneficial factors including prostaglandin E2, lipoxin A4 and nitric oxide. Besides the paracrine release, direct cell-cell transfer of these factors via extracellular vesicles extends the beneficial effects to the transfer of genetic material including DNA, mRNA and microRNA and of cell membrane components including cell surface receptors. Further important fields of action comprise their multiple antimicrobial activities, inhibition of epithelial-mesenchymal transition and lung fluid clearance. Besides the direct paracrine action, the transfer of vesicles and cell organelles via nanotube formation is a further important mechanism that amongst others enables cell energy stabilization via mitochondrial transfer. The initially described phenomenon of cell transdifferentiation and cell replacement at the site of injury is of only marginal relevance. This is underlined by the fact that exogenous MSC can only shortly engraft in the recipient lung [10,25,26].

In principle, MSC can be derived from nearly every human tissue but the most widely studied sources are bone marrow, peripheral blood, umbilical cord, Wharton’s jelly, placenta and adipose tissue. Recent advances in understanding the similarities and dissimilarities of MSC from different origin and the importance of aging on the functional properties of MSC have drawn the focus towards MSC obtained from the newborn infants’ umbilical cord and Wharton’s Jelly that possess superior anti-inflammatory and immunomodulatory functionality. Further frequently studied sources of MSC include bone marrow and adipose tissue [10].

## 3. Is the Therapeutic Efficacy of MSC to Prevent or Treat BPD in the Preclinical Setting Well-Founded?

First descriptions of the therapeutic potential of MSC for BPD emerged more than a decade ago. They demonstrated within rat rodent models that the deleterious effects of hyperoxia can be attenuated by exogenous MSC application [27,28]. Two subsequent pioneering studies published in 2010 paved the way towards the further evaluation of MSC to prevent or treat evolving BPD. Both studies were performed in the hyperoxia exposure rodent model and provided convincing evidence that classical injury patterns of BPD provoked by the immature lung’s inflammatory response were reverted including lung alveolar and vascular structures, right ventricular hypertrophy, pulmonary function and hemodynamics [29,30]. Already at these early stages, it came clear that the beneficial effects of MSC were not mainly executed by cell transdifferentiation but by the MSC secretome when MSC cell culture supernatants were included into the series of experiments [30]. Furthermore, it could be shown that therapeutic efficacy was achieved with both the intratracheal instillation and the intravenous injection administration routes. In summary, these two studies provided convincing evidence that exogenous MSC application has the therapeutic potential to overcome the deleterious consequences of lung resident MSC scarcity caused by hyperoxic exposure on further lung development.

The initial study results were in the meantime reproduced in overall 28 subsequent studies applying mesenchymal stem cells from different origin [27,28,31,32,33,34,35,36,37,38,39,40,41,42,43,44,45,46,47,48,49,50,51,52,53,54,55,56]. Study designs, key findings, follow-up investigations for safety and efficacy and details on origin, dosage, time point and route of MSC administration are presented within Table 2. Comparable results were obtained when lipopolysaccharide was applied mimicking the situation of amniotic infection or when hyperoxia was preceded by lipopolysaccharide exposure (for details see Table 3) [57,58,59]. Both prenatal and postnatal application of MSC proved therapeutic efficacy [57,58,59].

While the initial studies were designed to demonstrate the therapeutic potential of preventive MSC application, this is not feasible in the clinical setting. However, data on beneficial effects of MSC based therapies on alveolar and vascular structures and functional parameters during delayed application when BPD is already evolving opens the window for therapeutic interventions [60]. Examinations of the difference between initial and delayed MSC application yielded controversial results. One study systematically investigated early and late application during injury prevailed better results for the early and the early plus late intratracheal application of umbilical cord MSC to newborn rats exposed to hyperoxia while another study demonstrated beneficial effects for both early and rescue application [54,61]. A recent study on delayed MSC application after hyperoxic injury revealed that even intratracheal instillation in early adulthood and repeated retarded application in later adulthood have therapeutic potential within the rat model of hyperoxia. Interestingly, both alveolar and vascular structures were improved [62]. These data argue for reconsideration of the dogma that lung regeneration in BPD after initial injury is not possible. This is in line with findings that lung regeneration after MSC therapy was observed for other lung diseases like chronic obstructive pulmonary disease (COPD) as well. Therefore, these preliminary data and subsequent dissection of the underlying mechanisms can lay the basis for studies on MSC based therapies in severely affected former preterm infants beyond the NICU [63].

Studies on the optimal route of MSC administration confirmed the initial results of efficient delivery by intratracheal, intravenous or intraperitoneal injection (please refer to Table 2). Direct comparisons between application routes were performed only in one study for intratracheal versus intravenous and in one study for intratracheal versus intraperitoneal administration. Both demonstrated superiority of intratracheal application but equivalency of MSC dosages for different routes cannot be judged from these datasets [28,35]. In contrast, the intranasal route did not describe a therapeutic benefit when MSC were applied once but recently the repetitive application during and after a moderate injury proved therapeutic efficacy [32,48]. This allows the conclusion that the intranasal route is principally feasible but effects are more pronounced when direct routes are used. Beneficial effects were obtained mainly for umbilical cord and bone marrow derived MSC but three studies investigated placental tissue derived MSC and two studies amniotic fluid or amniotic membrane derived MSC with improvements in classical hallmarks of MSC action (compiled within Table 2) [47,52,64]. Only one study investigated MSC from different origin and reported MSC derived from the umbilical cord superior to MSC from adipose tissue (Table 4) [33]. Dose response studies demonstrated improved efficacy with increasing dosages up to 5 × 10^5^ cells given intratracheally [51]. Therapeutic efficacy was similar for preventive and rescue therapy at a dosage of 3 to 6 × 10^5^ cells given intratracheally per animal. Of note, improvements in lung structures persisted into adulthood and the systematic review confirmed efficacy in mice and rats within a dose range from 5 × 10^4^ to 5 × 10^6^ cells per animal [61,65].

Gender specific differences need to be carefully monitored in the future, as in the rat hyperoxia model, MSC derived from female donors’ bone marrow displayed improved reduction of vascular remodeling in male recipients [66]. Importantly, results using MSC of human and rodent origin displayed comparable benefits that are detailed within Table 2. It needs to be taken into account that the presence of exogenous MSC is restricted to a short time interval after application that was confirmed in one study on BPD that evaluated MSC presence in more detail [32]. That might argue towards repeated applications to improve therapeutic efficacy.

The preclinical pathomechanistic studies give a robust overview about the main actions of exogenous MSC application in the BPD models of mice and rats: MSC application at least partially or completely reverts the inflammatory response and cytokine disbalance in the lung evoked by hyperoxia. MSC application attenuates the increase in pro-inflammatory cytokines like IL-1α, IL-1β, IL6, IF-γ, CCL5, CXCL7, MIP-1α, MIP-2, TNF-α, TGF-β1, factors like CTGF, inhibitors like TIMP1 and cell adhesion molecules like L-selectin (CD62L) and sICAM-1 (CD54) while the level of anti-inflammatory cytokine IL-10 and growth factors angiopoietin-1, VEGFA, HGF and PECAM is retained. As a consequence, recruitment of pro-inflammatory M1 macrophages and neutrophils, myeloperoxidase activation and oxidative stress to the lung is dampened while retention of macrophages with an anti-inflammatory, lung resident M2 phenotype is preserved [28,31,33,34,35,36,37,38,39,40,42,43,44,46,49,50,51,52,53,54,55,56,57,67]. The beneficial effects on the intrapulmonary cytokine balance account for the documented attenuation of inflammatory cell influx to the site of injury, mainly macrophages and neutrophils, the reduced release of proteases and the attenuation of the disequilibrium of pro- and anti-apoptotic Bcl-2 family member expression and attenuated apoptosis induction in the lung [28,29,32,67]. Decorin and pentraxin-related protein PTX3/tumor necrosis factor-inducible gene 14 protein (PTX3) were identified as critical drivers towards M2 macrophage polarization, which was associated with reduced inflammatory cytokine release and a better-preserved lung structure [46,49]. The inhibition of upregulation of formyl peptide receptor-1 (FPR-1) which is known for its sensor function in inflammation is one further finding that accounts for MSC function as MSC transplantation to wildtype mice exposed to hyperoxia was as efficient as FPR-1 knockout [43]. The attenuation of pro-fibrotic TGF-β1 is accompanied by decreased collagen 1 and aberrant elastin deposition and reduced lung elastase activation [31,41,51]. Looking in detail at further aspects important for proper lung function, exogenous MSC application was associated with better preserved alveolar type II cell counts and aquaporin-5 channel expression that is responsible for fluid secretion [67]. Besides the potent inhibition of inflammation, MSC application was associated with suppression of sonic hedgehog pathway signaling and of hyperoxia-induced activation of the renin-angiotensin system in the lung [40,47]. In one study, exogenous MSC application even increased the number of bronchioloalveolar stem cells extending their beneficial effects to the lung stem cell progenitor pool [68].

Although not all hallmarks of MSC action described for lung injury across ages have been reproduced in BPD models, it comes clear that exogenous MSC act via the identical key mechanisms described in other lung diseases and beyond [10]. The studies in rodents were systematically reviewed in the meantime and came to the conclusion that MSC have a robust and overall positive effect on the pulmonary outcome. Beneficial effects were stated for initial and delayed MSC application and different sources, dosages and routes of exogenous MSC application as detailed above [65,69]. Benefits of MSC therapy were not restricted to the lung but also included reduced brain injury among others even when applied topically to the lung. The main mechanism of MSC action was again attributed to the attenuation of inflammation and cell death in the brain [37]. Therefore, it is intriguing to look beyond BPD to incorporate results and considerations from other disease entities to bring the MSC strategy as quickly as possible to clinical success. Some preliminary results indicate that only specific subpopulations of MSC account for their beneficial effects. Therefore, a further research focus that investigates differences between MSC subpopulations needs to be established [70,71]. Most investigations comparing freshly isolated and thawed cell products demonstrated optimal results for fresh MSC preparations but one study did not detect any difference arguing towards a detailed investigation in future BPD studies as the use of deep-frozen cell products eases the application [72]. In some studies, the retention of sparce MSC in the lung has been described for even prolonged periods of time issuing questions to the long-term safety of this approach. Although no long-term side effects of infectious complications, therapy associated deaths or malignancies were observed following MSC application so far [55,73,74], safety concerns need to be carefully monitored as pointed out recently [75]. Consequently, we will discuss the results from cell-free MSC based strategies and their advantages in a later chapter but previously, we will take a look at available clinical trial data.

**Table 2 ijms-22-01138-t002:** Therapeutic actions of mesenchymal stem cells (MSC) to prevent or treat BPD in the hyperoxia rodent model.

Experimental Lung Disease Model	Species	Cell Source	MSC Species	Dose (Cells)	Application Route	Time Point of Application	Effect on Survival	Histologic Evaluation	Functional Properties	Molecular Changes	Reference
hyperoxia 90%, P1–P5	mice	human	BMMSC	2.5 × 10^5^	i.t.	P5			attenuation of M1 macrophages, collagen deposition, retention of M2 macrophages		[44]
hyperoxia 80%, P1–P14	mice	human	UCBMSC	2 × 10^5^	i.t.	P5	no effect (100% survival in all groups)	attenuation of alveolar and vascular lung pathology	reduction of formyl peptide receptor-1 expression and similar effects on apoptosis, VEGFA levels and influx of macrophages and neutrophils as in FPR-1 knockout mice		[43]
hyperoxia 90%, P1–P7	mice	human	UCTMSC	1 × 10^5^ or 5 × 10^5^ or 1 × 10^6^	i.n. or i.p.	P5		reduced alveolar remodeling	normalization of lung function parameters		[32]
hyperoxia 75%, P1–P14	mice	mice	BMMSC	5 × 10^4^	i.v.	P4		preserved alveolar and vascular structures		reduced influx of macrophages/neutrophils	[29]
hyperoxia 60%, P1–P45	mice	mice	BMMSC	1 × 10^5^	i.p.	P7	increased survival	better preserved alveolar structures, attenuated fibrosis		inhibition of IL-1β, TNF-α, TGF-β1 upregulation	[31]
hyperoxia 60%, P1–P14	mice	mice	BMMSC	1 × 10^6^	i.v.	P1		reduced alveolar hypoplasia		better preserved VEGFA, reduced TGFβ1	[34]
hyperoxia 60%, P1–P14	mice	mice	BMMSC	1 × 10^6^	i.v.	P1 and P7		improved airway structures		improved PECAM and VEGFA, reduced MMP-9	[36]
hyperoxia 60%, P1–P14	rat	human	AFMSC	1.5 × 10^6^	i.t.	P21	100% survival in all groups	better preserved alveolar and vascular structures		reduced IL1β, IL6, IF-g, TGF-β1, apoptosis induction, preserved VEGFA	[52]
hyperoxia 95%, P1–P14	rat	human	BMMSC	3 × 10^5^ (P4) or 6 × 10^5^ (P14)	i.t.	P4 or P14		attenuated alveolar and vascular changes by P4/P14	improved exercise capacity		[61]
hyperoxia 85%, P1–P14	rat	human	PTMSC	1 × 10^5^	i.t.	P5	no effect	reduced alveolar hypoplasia		reduced apoptosis, IL1β, MIP-2	[39]
hyperoxia 85%, P1–P14	rat	human	PTMSC	1 × 10^5^	i.t.	P5	no effect	attenuation of alveolar rarefication		reduced IL6, TNF-α, activation of the renin-angiotensin system	[40]
hyperoxia 85%, P4–P15	rat	human	PTMSC	9 × 10^5^	i.v.	P15	no effect	improved alveolarization, vascularization	associated with suppression of sonic hedgehog signaling		[47]
hyperoxia 80%, P1–P14	rat	human	PTMSC	1 × 10^6^	i.t.	P7	increased survival	amelioration of lung injury			[64]
hyperoxia 90%, P1–P14; 60%, P15–P21	rat	human	UCBMSC	5 × 10^5^	i.t.	P3 and/or P10	increased survival for treatment P3/P3 + P10	better preserved alveolar structures for P3/P3 + P10		reduced oxidative stress, TNF-α, IL-1β, IL6, TGF-β, TIMP1, CXCL7, RANTES, L-selectin, sICAM-1, better preserved HGF, VEGFA for P3/P3 + P10	[54]
hyperoxia 95%, P1–P14	rat	human	UCBMSC	5 × 10^3^ or 5 × 10^4^ or 5 × 10^5^	i.t.	P5	increased survival in medium and high MSC dosage intervention groups	dose-response relationship of attenuation of lung injury		dose response relationship for attenuation of myeloperoxidase activity, TNF-α, IL1β, IL6, TGF-β and oxidative stress	[51]
hyperoxia 90%, P1–P14	rat	human	UCBMSC	1 × 10^5^	i.t.	P5	increased survival	reduction of lung injury	increased M2 macrophages IL10, reduced M1 macrophages, IL6, IL8		[49]
hyperoxia 90%, P1–P14	rat	human	UCBMSC	1 × 10^5^	i.t.	P5	increased survival	MSC with high decorin expression better preserved alveolar structures	MSC with high decorin expression inhibited IL6, IL8, retained IL10 decorin responsible for M2 macrophage polarization		[46]
hyperoxia 90%, P1–P14	rat	human	UCBMSC; ATMSC	5 × 10^5^	i.t.	P5	no effect	reduced alveolar hypoplasia			[33]
hyperoxia 90%, P1–P14	rat	human	UCBMSC	5 × 10^5^ (i.t.) or 2 × 10^6^ (i.v.)	i.t. or i.v.	P5	increased survival	reduced alveolar hypoplasia		reduced macrophages, i.t. additional inhibition of apoptosis, MIP1α, TNF-α, IL6, CTGF, better preserved VEGFA, HGF	[35]
hyperoxia 90%, P1–P14	rat	human	UCBMSC	5 × 10^5^	i.t.	P5	hyperoxia lower than normoxia, hyperoxia + MSC similar to normoxia	improved airway and brain structures		reduced IL-1α, IL-1β, IL6, TNF-α, better preserved VEGFA	[37]
hyperoxia 90%, P1–P14	rat	human	UCBMSC	5 × 10^5^	i.t.	P5		benefits for alveolarization, angiogenesis		beneficial effect on cell death, activated macrophages, IL1α, IL1β, IL6, TNF-α	[42]
hyperoxia 90%, P1–P14	rat	human	UCBMSC	5 × 10^5^	i.t.	P5	no effect	improved alveolar and vascular structures	reduced neutrophils/macrophages, inflammatory foci		[55]
hyperoxia 90%, P1–P14	rat	human	UCBMSC	5 × 10^5^	i.t.	P5		benefits on alveologenesis and vasculogenesis		reduced apoptosis, macrophages, IL1α, IL1β, IL6, TNFα	[56]
hyperoxia 95%, P1–P14	rat	human	UCBMSC	2 × 10^6^ i.t. or 5 × 10^5^ i.p.	i.t. or i.p.	P5	not significantly improved	i.t. only: preservation of alveolar structures		reduced apoptotis, myeloperoxidase activity and IL6 level; i.t. only: attenuated TNF-α, TGF-β1, α-SMA expression, collagen deposition	[28]
hyperoxia 80%, P1–P21	rat	human	UCTMSC	3 × 10^5^	i.t.	P7	increased survival	attenuation of lung alterations		reduced elastase activity, aberrant elastin deposition, TGF-β1	[41]
hyperoxia 60%, P4–P7	rat	human	UCTMSC	5 × 10^5^	i.n.	P4, P10 and P20	no effect	improved alveolarization and vascularization	associated with gene regulation for angiogenesis, immunomodulation, wound healing, cell survival		[48]
hyperoxia 60%, P1–P14	rat	human	UCTMSC	in total 6 × 10^6^	i.t.	P3, P7 and P10	no effect (100% survival in all groups)	retention of alveolarization, vascularization			[45]
hyperoxia 95%, P3–P10	rat	rat	BMMSC	5 × 10^4^	i.v.			preserved alveolar structures		attenuation of TGFβ, TNF-α upregulation	[27]
hyperoxia 95%, P1–P14	rat	rat	BMMSC	1 × 10^5^	i.t.	P4 or P14	increased survival	preserved alveolar and vascular structures	reduced pulmonary hypertension, improved exercise tolerance		[30]
hyperoxia 80%, P1–P15	rat	rat	BMMSC	1 × 10^5^	i.v.	P5	no effect	improved alveolar and vascular structures	reduced pulmonary hypertension	reduced M1 macrophages, IL6	[76]
hyperoxia 95%, P3–P10	rat	rat	BMMSC	1 × 10^5^	i.v.	P10		attenuated lung injury		suppression of TNF-α, TGF-β upregulation	[50]
hyperoxia 95%, P3–P10	rat	rat	BMMSC	1 × 10^5^	i.v.	P10	increased survival			preservation of VEGFA, AQP5, SPC expression	[67]
hyperoxia 90%, P2–P16	rat	rat	BMMSC	2 × 10^6^	i.t.	P9		acute and long-term improvements in alveolar and vascular development		reduced IL1β and IL6 upregulation, preserved Ang-1 and VEGFA	[53]
hyperoxia 85–90%, P2–P21	rat	rat	BMMSC	1 × 10^6^	i.t.	P7		improved airway and vasculature structures			[66]
hyperoxia 85%, P1–P21	rat	rat	BMMSC	1 × 10^6^	i.t.	P7		improved alveolarization and angiogenesis		reduced influx of inflammatory macrophages, neutrophils, reduced IL-1β, improved IL-10	[38]

MSC—mesenchymal stem cells; BPD—bronchopulmonary dysplasia; Pn—postnatal day n; BMMSC—bone marrow mesenchymal stem cells; UCBMSC—umbilical cord blood mesenchymal stem cells; UCTMSC—umbilical cord tissue mesenchymal stem cells; AFMSC—amniotic fluid mesenchymal stem cells; ATMSC—adipose tissue mesenchymal stem cells; PTMSC—placental tissue mesenchymal stem cells; i.v.—intravenous; i.t.—intratracheal; i.p.—intraperitoneal; i.n.—intranasal; IL—interleukin; TNF—tumor necrosis factor; IF—interferon; TGF—transforming growth factor; VEGFA—vascular endothelial growth factor; SMA—smooth muscle actin; Ang—angiopoietin; TIMP—tissue inhibitor of metalloproteinases; CXCL—chemokine (C-X-C motif) ligand; RANTES—regulated upon activation normal T cell expressed and secreted; sICAM-1—soluble intercellular adhesion molecule; HGF—hepatocyte growth factor; MIP—macrophage inflammatory protein; CTGF—connective tissue growth factor; PECAM—platelet endothelial cell adhesion molecule; MMP—matrix metalloproteinase; FPR—formyl peptide receptor.

**Table 3 ijms-22-01138-t003:** Therapeutic actions of MSC to prevent or treat BPD in the LPS-induced rodent model.

Experimental Lung Disease Model	Species	Cell Source	MSC Species	Dose (Cells)	Application Route	Time Point of Application	Effect on Survival	Histologic Evaluation	Functional Properties	Molecular Changes	Reference
intrauterine LPS	mice	mice	BMMSC	2 × 10^6^	i.a.	G17		better preserved lung maturation		ErbB4 required for MSC action	[59]
intrauterine LPS plus hyperoxia 85%, P1–P14	rat	human	PTMSC	3 × 10^5^ or 1 × 10^6^	i.t.	P5	increased survival from P6 to P9, no difference at P14	reduced alveolar and vascular hypoplasia,		inhibition of TNF-α, IL6, CTGF, collagen density, better preserved VEGFA	[57]
intrauterine LPS	rat	human	PTMSC	3 × 10^5^ or 1 × 10^6^	i.t.	P5		reduced alveolar and vascular hypoplasia		reduced influx of inflammatory macrophages	[58]

MSC—mesenchymal stem cells; BPD—bronchopulmonary dysplasia; Pn—postnatal day n; Gn—gestational day n; BMMSC—bone marrow mesenchymal stem cells; PTMSC—placental tissue mesenchymal stem cells; i.t. —intratracheal; i.a. —intraamniotic; LPS—lipopolysaccharide; IL—interleukin; TNF—tumor necrosis factor; VEGFA—vascular endothelial growth factor; CTGF—connective tissue growth factor.

**Table 4 ijms-22-01138-t004:** Comparison of different stem cell preparations in BPD animal models.

Experimental Lung Disease Model	Species	Cell Source	MSC Species	Dose (Cells)	Application Route	Time Point of Application	Effect on Survival	Histologic Evaluation	Functional Properties	Molecular Changes	Reference
hyperoxia 90%, P1–P14	rat	human	UCBMSC; ATMSC	5 × 10^5^	i.t.	P5	no effect on survival	better preserved alveolar structures by UCBMSC than ATMSC; improved angiogenesis only after UCBMSC		decreased cell death, macrophage influx, inflammatory cytokine levels only after UCBMSC	[33]

MSC—mesenchymal stem cells; BPD—bronchopulmonary dysplasia; Pn—postnatal day n; ATMSC—adipose tissue mesenchymal stem cells; UCBMSC—umbilical cord blood mesenchymal stem cells; i.t.—intratracheal.

## 4. Do the Results from MSC Application within First Phase I Clinical Trials Justify to Further Pursue This Approach?

The highly promising results from most preclinical studies in rodents raised great hopes that allogenic MSC therapy can effectively alleviate the consequences for the lung following preterm birth. The first phase I study performed in Korea was performed in n = 9 extremely low birth weight infants that required ventilator support beyond postnatal day 5 of life for respiratory insufficiency. The first three infants received a dosage of 1 × 10^7^ human umbilical cord derived MSC per kilogram body weight once intratracheally, the further six infants 2 × 10^7^. Of relevance, the MSC preparation went through a freeze-thaw cycle before application. Cell application and clinical follow-up revealed no acute toxicities or side effects. The combined analysis of all nine infants demonstrated statistically significant differences in the pulmonary outcome when the BPD criterion at 36 weeks of gestation was used. While 3/9 treated infants fulfilled the criterion moderate or severe BPD, there were 13/18 infants in the historic control group [77]. The reduced BPD rate in cases was further substantiated by the attenuation of the lung’s proinflammatory cytokine response. Studies on levels of the characteristic pro-inflammatory cytokines IL6, IL8 and TNF-α and on matrix metalloproteinase 9 revealed an attenuation in tracheal aspirates on day 7 after MSC administration. In line, the respiratory severity score tended towards lower values than in controls (*p* = 0.05) [77]. Results from the follow-up of treated infants are currently available until the age of 2 years. Persistent beneficial effects with respect to oxygen supply at home, rehospitalizations for pulmonary reasons and somatic growth were confirmed. No adverse effects were detected on the psychomotor outcome [78]. These results raise great hopes that MSC application is an efficient approach in the future. A subsequent phase II study by the same authors completed recruitment already, we are awaiting publication of results from this trial (NCT01828957) and from the ongoing follow-up. A second early intervention phase I study from the US investigated the identical MSC preparation obtained from the human umbilical cord given to 12 preterm infants with a birth weight <1000 g and gestational age <28 weeks once on day 6–14 of life including a dose escalation step. Cell numbers applied were identical to the initially published study. No randomization and no historic cohort were included. The intratracheal application was well tolerated without any predefined serious adverse events of the cardiorespiratory system, anaphylactic reactions or deaths recorded. Ten out of the twelve treated infants developed severe BPD, two infants mild BPD. Severe affection of the lung was further testified by a median duration of mechanical ventilation of 35 days, a median of 114 days on oxygen supply and corticosteroid use in 8 of these infants. Of notice, one death on day 161 of life after end of study observation due to pulmonary hypertension and 3 sepsis events were recorded that argues towards careful surveillance in any future study [79]. Compared to the initial study, infants were more immature and displayed more severe respiratory disease that might account for differences in outcome results.

One further single center open-label phase I study investigated the safety of late allogenic amnion cell transplantation to severely affected infants with established BPD requiring invasive ventilation or non-invasive respiratory support with oxygen fractions of 0.3–0.5. Cells were given intravenously with a dosage of 1 × 10^6^ per kilogram body weight. The first infant showed cardiorespiratory compromise that was traced back to potential pulmonary embolism by the infusion. After changing the administration technique including further cell dilution and inserting an in-line filter infusions were tolerated without adverse events in the other five infants. No side effects were observed during follow-up in the ward and the death in one child one month after infusion was ascribed non-related to the intervention. Within the first week following the infusion, no improvements in respiratory support were observed but in 3 infants FiO_2_ requirements decreased [80]. The follow-up until the age of 24 months did not reveal any therapy-associated side effects although validity remains restricted in a population of severely affected infants with ongoing problems of somatic growth and cardiorespiratory and psychomotor function [81]. The subsequent phase I dose escalation study increasing the dosage up to 3 × 10^7^ cells per kilogram body weight commenced already with results including the 24 month follow-up expected for 2022. While the initial study aimed to address only safety issues, the authors now based the dose-escalation on data derived from studies in humans and animals that proved therapeutic efficacy [80,81,82]. For a comprehensive list of further ongoing phase I and II MSC trials, we refer to this recently published reference [83].

All three published studies to date have in common that they were designed to monitor safety aspects but not benefits for the pulmonary outcome. Reliable responses to the questions of optimum source and preparation, dosage and route of application are a prerequisite for stepping towards the next level. To speed this process up, the look at MSC trials for other pulmonary diseases is intriguing as lung diseases across entities and ages have several key pathomechanisms in common. Shortly summarizing, none of these more than twenty trials prevailed any safety concerns. Most studies used the intravenous route of application and the most frequently applied dosage was 1x10^6^ cells per kilogram body weight. Outcomes were heterogenous: While some studies did not detect any benefit, some studies displayed the expected reduction in markers of inflammation and others demonstrated short- or longer-term beneficial effects for lung function or personal well-being [10]. As for preterm infants, studies in adults mostly included severely affected patients. That might conceal the therapeutic potential as these patients have established disease and inflammatory processes. Interpretation of the available results is hampered by the heterogeneity of study outlines and different origin, preparation techniques, dosages, timing and administration routes. Results from the first BPD studies are confirmed by the summary of results from adult trials where significant beneficial effects on the lung became only visible when applied MSC dosages exceeded 1 × 10^6^ cells per kilogram body weight or when repeated infusions were administered [10,77,84]. The timing of MSC application remains a critical issue despite the described benefits of delayed application in animal trials. Early application before aggravation of lung injury might pave the way to prove efficacy. Safety concerns need to be critically monitored within all ongoing and future studies that are discussed in the next chapter and need to include sepsis and pulmonary hypertension. Besides the presumed potential side effects including drug-drug interactions of MSC and surfactant need to be closely monitored [39].

## 5. Is MSC Application Safe?

A recent systematic review and meta-analysis on MSC therapy safety concluded that no safety concerns are evident [73]. Despite, it remains an unresolved question whether MSC can be applied safely to the immunocompromised preterm infant. Although plenty of studies investigated persistence of the MSC in the lung, preclinical studies did mostly not detect permanent engraftment and most publications described disappearance within a few days. There still remains uncertainty whether the lack of MSC persistence was due to inappropriate techniques. Concerns are fostered by the fact that MSC possess the capability to adapt their immunologic function enabling them to escape immunologic detection and cell control. In theory, this can result in uncontrolled cell proliferation and malignancy. Furthermore, MSC can release a plenty of pro-inflammatory cytokines like interleukins and macrophage stimulating factors to activate host defense mechanisms against infection and injury [26,85]. Special focus needs to be directed towards the release of TGF-β and its immunomodulatory and pro-fibrotic functions [86]. Whether this is a real obstacle to therapy success needs further studies. The published results hint towards a distinct reaction of lung resident MSC and bone marrow derived MSC but this needs further confirmation [87]. The data from the amnion cell transplantation study confirm that pulmonary embolism after MSC application is an ever-present concern that needs to be considered in any study outline [80]. Adequate dilution, slower infusion rates, in-line filtration and anticoagulation represent therapeutic options to resolve this issue. Furthermore, the timing of MSC application needs critical review. A probably optimal efficacy of the intervention needs to be weighed against the therapeutic aim to restrict this new therapy to infants with pulmonary sequelae that requires delayed application. Although the late administration of amniotic cells did not prevail any adverse effects, the therapeutic efficacy might be restricted as inflammatory damage to the immature lung has already been executed. A solution to this is the early identification of infants at high risk for BPD by biomarker approaches to provide MSC early on [88,89].

Besides these impeding safety concerns of MSC application, there still remain a plenty of unresolved issues about the optimal source and stability of cell preparations [90]. It is well known that during passaging, MSC undergo molecular changes and experience phenotype changes with increasing donor age that impair therapeutic efficacy [91,92]. Therefore, the continuous availability of freshly isolated MSC and harmonization of cell product preparation remain immanent issues [93]. Frozen MSC preparations were studied in BPD but results from other disease entities clearly indicate that this approach will not provide identical therapeutic efficiency. Furthermore, regular supply to the premature infant requires the provision of large scaled MSC preparations compared to the low amount required for studies in the rodent model. These capacities are not available and need to be build up before performing any larger scale clinical study. Therefore, alternative strategies with the aim of having a stable product in sufficient quantities readily available off the shelf are required and therefore discussed within the next chapter.

## 6. Is the Secretome the Key to Practicality and Safety of MSC Application?

As discussed in detail before, the main action of MSC is via their secretome. These so-called extracellular vesicles (EV) contain many factors of MSC that account for beneficial effects on inflammation, organ development and repair. Besides cytokines and growth factors, EV contain gene products, mRNAs and microRNAs [94]. Lower immunogenicity and smaller size with higher ease in crossing of biological barriers compared to MSC make them especially attractive for BPD prevention and treatment [95]. Today, EV and cell culture supernatants from MSC called conditioned media demonstrated therapeutic efficacy in 14 published preclinical studies in rodent models of BPD that are summarized in Table 5. Already at an early stage of research, it was demonstrated that EV have comparable therapeutic efficacy on alveolar and vascular development and exercise capacity [29]. The repeatedly described short presence and low engraftment rates of MSC deliver the reason for equal potency. Follow-up studies ensured that there was no long-term disadvantage compared to MSC application [61]. Application during the acute phase of injury but even after 14 days of hyperoxic exposure proved efficient to attenuate or completely revert the deleterious consequences for lung structure and functional properties [29,45,60,61,68,96,97,98,99,100,101]. Detailed structural and functional analyses demonstrated benefits not only for pulmonary hypertension but for peripheral vascular remodeling and for the pool of bronchioloalveolar stem cells as well [60,68,96,97,100]. Dissection of the inflammatory response revealed a reduced influx of neutrophils, an M1 to M2 shift in macrophages, suppression of pro-inflammatory cytokines and augmentation of anti-inflammatory cytokines and growth factors [53,97,99]. These actions observed for BPD animal models are in line with the main actions described for EV in general [102]. In the context of EV delivery to the immature lung, further drivers of BPD as infections and microbial dysbiosis have not been evaluated so far [103,104]. However, promising studies have been conducted on the treatment of neonatal sepsis [105]. Lastly, the so far not studied EV properties of stabilizing the energy balance of target cells and direct antibacterial activities have the potential to demonstrate further therapeutic efficacies and to extend the indications for EV application to the preterm lung [106,107].

The incorporation of EV is not cell type specific and EV were verified in type II cells, in lung fibroblasts and pericytes when given intratracheally [42]. Building the bridge into the clinics, studies included EV of clinical grade quality [45]. As for the studies on MSC from different origin, EV from umbilical cord and bone marrow MSC demonstrated equal potency [97]. Direct comparison of exosomes and MSC was executed with human and rodent derived MSC from umbilical cord and bone marrow. There was no obvious discrepancy in therapeutic efficacy between cells from both origins [60,68,97,99]. In one experimental setting, exosome delivery better preserved alveolar and vascular development in animals exposed to hyperoxia but the outcome measures did not reach statistical significance. Two other settings were unable to display differences in outcome [45,53,61]. One study using EV derived from adipose tissue MSC revealed reduced efficacy compared to the direct cell application that argues towards an in-depth comparison of preparations before using them in clinical settings [64]. These data suggest a thorough evaluation of EV potency needs to be conducted when comparing different treatment approaches. This requires the harmonization of EV production, purification, storage, quantification and the establishment of standardized potency assays [108]. EV potency does not only depend on the EV content of cytokines and cytoplasmic components but also on their surface biomarker and receptor expression [94,109]. The former obstacles of having high amounts of EV available for the conduction of clinical studies in the preterm infant, are in the meantime overcome by the rapid increase in companies stepping into the field of EV production and provision for clinical trials [110]. Adequate and uniform distribution of EV in the lung after intravenous or intratracheal application need to be verified. At least for the i.v. route, close monitoring of successful distribution in the injured lung is a prerogative for therapy success [111]. Further aspects include the optimization of the first application time point and required repetition of EV administration. Safety issues need to be monitored precisely as the transfer of genetic material, cytokines and growth factors might as well cause undesirable effects including malignancy transformation. But the available results from the rodent hyperoxia models argue towards focusing efforts on EV therapy as the next level. One phase I study of EV therapy to prevent BPD registered at the NIH clinicaltrials.gov homepage (NCT03857841) is currently recruiting infants. Results will be available by the end of 2021 and might shed light on the safety of the EV approach in the preterm infant. Taking a look at EV therapy in adult patients across disease entities, results displayed some beneficial effects but the great enthusiasm about this new therapeutic strategy arisen from preclinical models still awaits further confirmation [95,102]. For the time being, BPD seems to be one special candidate for EV therapy within the many approaches to lung diseases [112]. Reviewing the next steps to further improve therapeutic efficacy is the aim of the following and last section.

## 7. Is Cell Engineering the Ultimate Step to Therapy Success?

Although the discrepancies between the consistently beneficial effects in the rodent BPD models and heterogenous results in preterm infants can be partly explained by differences in dosing, the heterogeneity of BPD severity in participating patients and the approach of clinical studies to demonstrate safety but not efficacy, the further improvement of efficacy is the next necessary step in MSC research. We will review the available data on MSC and EV together due to the scarce data available for the BPD setting. In principle, strategies can be separated into application of MSC together with a further therapeutic (Table 6) and biochemical (Table 7) or genetic (Table 8) modification of MSC before therapeutic application. The combinatorial approach of MSC plus recombinant erythropoietin was the first studied in the hyperoxia BPD model. All readout parameters displayed further improvements for the combinatorial approach including alveolar lung structures, better preserved VEGFA and reduced MMP-9 activation [36]. While erythropoietin augmented the efficacy of exogenous MSC, this was not observed for MSC plus surfactant. While each treatment alone attenuated the hyperoxia induced alveolar hypoplasia, no additional benefit was detected for the combination [39]. MSC preconditioning has also been tested in the experimental setting of BPD. One study performed in the initial phase of BPD research constituted that conditioned medium harvested from bone marrow derived MSC exposed to hyperoxia proved superior efficacy with respect to alveolar structures and pulmonary hypertension than naïve MSC [96]. These data need reproduction as mostly hypoxic but not hyperoxic preconditioning improves MSC functionality [113]. Gene modification studies are suited to describe the therapeutic relevance of a specific factor in the prevention of BPD pathology with the vision of engineering MSC or EV to gain the best possible treatment results. For TSG-6, which is known for its capacity to modulate amongst other actions macrophage plasticity towards an anti-inflammatory phenotype, knockdown in human umbilical cord derived MSC exosomes markedly attenuated the beneficial effects on lung and heart [99]. Knockdown of VEGFA nearly completely abrogated the beneficial effects of MSC and exosome application in two independent studies [42,56]. In this way TSG-6 and VEGFA were ascribed a dominant role in MSC and EV activity and reverse strategies of MSC transduction will aim to improve the activity of these two proteins. Transduction of stem cells from the amniotic fluid for VEGFA augmented all key features of BPD while naïve cells only improved inflammation and vascular development confirming a central role of VEGFA across stem cell entities [114]. Similarly, knockdown of stromal-derived factor-1 (SDF-1) in MSC before transplantation partially abrogated the beneficial anti-inflammatory and pro-angiogenic activity attributing SDF-1 an important function during MSC action that was recapitulated in preterm infant lung autopsy studies [38,115]. For decorin and PTX3, their action was ascribed to modulation of macrophage function towards an M2 status using siRNA experiments and thereby better preserving lung development [46,49]. In line, MSC transduction with 7ND-CCL2, a potent CCR2 antagonist, proved efficacy to prevent hyperoxia induced distortion of alveolar and vascular development in the immature lung which was associated with the reduced influx of M1 macrophages and pro-inflammatory cytokine expression [76].

MSC engineering has been evaluated much more detailed in other lung diseases, mostly acute lung injury. The therapeutic efficacy has been demonstrated for a plenty of growth factors and anti-inflammatory cytokines that stipulated the resolution of inflammation, lung repair and regeneration. Cell surface receptors constitute another suitable target to improve MSC recruitment to the site of injury [10]. Overall, these results raise great hopes to make MSC based therapies more efficient and safer in the near future.

## 8. Concluding Remarks

Complex interactions between inflammatory injury and disruption of physiologic lung development represent key features of BPD. Therefore, it is inspiring to search for broad-acting therapeutics. The MSC-based approach is especially attractive as it combines anti-inflammatory and growth and repair promoting properties. It has finally the potential to add a powerful strategy to the short list of available medications to prevent BPD. Although it is much too early to judge the ultimate efficacy in preterm infants, the congruent results obtained so far exclusively in rodent models are promising. Of course, the criticism of studies in mouse models is justified as none of the actual therapeutics evolved from studies in mice [116]. As for all other new therapies, a careful and complete evaluation of MSC-based approaches covering the lung microenvironment and the comprehensive documentation of side effects under particular conditions like the recently described increased pulmonary artery embolism in a sheep extracorporeal membrane oxygenation model of acute respiratory distress syndrome is indispensable [117,118]. On the other hand, even a much lower efficacy in preterm infants can alleviate the tremendous disease burden of BPD where the best available medical approaches need a number of 10 treated infants to prevent one BPD case [12,20]. And even preterm infants not fulfilling the BPD criterion have relevant lung function restrictions persisting throughout life [5]. Within the established medications to prevent BPD, corticosteroids display the highest therapeutic efficacy. This is not surprising as they are used as anti-inflammatory agents. But an important side effect is often neglected in the situation where successful extubation or even the survival of the child depends on their application. Corticosteroids are powerful agents that disrupt further lung development [119]. With the MSC approach, two birds can be tackled with one stone. The next steps to a successful introduction of MSC-based therapies into the clinics include the standardization of experimental approaches with respect to dosing, timing and selection of precisely defined patient groups and the production and efficacy testing of cell preparations. Ongoing investigations within the baboon model of borderline viability are suitable to confirm the results from rodents and can help to determine functional improvements in more detail than mouse or rat models [120]. For these reasons, the just published meta-analysis was not able to provide final conclusions [69]. The robust comparison of different therapeutic approaches and the standardization of experimental settings are necessary prerogatives to dissolve these unanswered questions. Special focus needs to be drawn to detect unexpected adverse and inter-species effects [121]. The studies from recent years provided convincing evidence that inflammatory and lung growth promoting pathways use common signaling pathways. Thereby, the overshooting on the one hand but the complete interruption on the other hand might both end up with aggravated lung injury what has been mostly detailed for NFκB [7,122,123,124]. Lastly, inflammation in the preterm infant is a systemic disease that is not limited to the lung. Other important outcomes arising from inflammation include brain injury, necrotizing enterocolitis or retinopathy of prematurity. It is not surprising that the rare studies on further organs beside the lung provide convincing evidence that systemic MSC based therapies alleviate these injuries [37]. With this in mind, the focus of research needs to be expanded. A crucial moment is reached to join experiences, research efforts and results to translate this promising therapy successfully into the clinics. The high efficacy of MSC under optimized laboratory conditions where the setting is designed to dissect effects does not reflect the complex and multifactorial reality in the clinics. The efforts of MSC engineering designed to strengthen the efficacy and to abrogate side effects need to be followed closely and are of high therapeutic potential.

It is intriguing that a one-time application shall prevent or cure BPD. This approach was probably preferred due to the high costs of MSC and EV production. But the therapy is applied in an inflamed milieu where the causes of lung injury get not immediately abrogated by the application per se but the infant still depends on ventilation and oxygen supply. These facts together with the only short-term presence of MSC in the lung argue towards repetitive or permanent application as standard of care that might provide the ultimate key to cure BPD.

## Figures and Tables

**Table 1 ijms-22-01138-t001:** Drugs with proven therapeutic efficacy to prevent bronchopulmonary dysplasia (BPD).

Drug	Intervention	Control	Odds Ratio	Number Needed to Treat	Reference
surfactant ^1^	437/805	517/791	0.83	9	[17]
	(54.3%)	(65.4%)	(0.77–0.90)		
vitamin A	486/1000	546/1000	0.87	11	[16]
	(48.6%)	(54.6%)	(0.77–0.99)		
caffeine	350/963	447/954	0.64	9.5	[12,13,14]
	(36.3%)	(46.9%)	(0.52–0.78)		
azithromycin	81/161	90/149	0.83	10	[15]
	(50.3%)	(60.4%)	(0.71–0.97)		
corticosteroids	498/1964	633/1965	0.79	n.a.	[18]
<8 days	(25.4%)	(32.3%)	(0.72–0.87)		
corticosteroids	143/295	183/285	0.77	n.a.	[19]
>7 days	(48.5%)	(64.2%)	(0.67–0.88)		

^1^ Only significant for the combined outcome of BPD or death. BPD—bronchopulmonary dysplasia; n.a.—not available.

**Table 5 ijms-22-01138-t005:** Efficacy of MSC derived therapeutics (extracellular vesicles/conditioned media) in BPD rodent models.

Experimental Lung Disease Model	Species	Cell Source	MSC Species	MSC Preparation	Dose (Cells)	Application Route	Time Point of Application	Histologic Evaluation	Functional Properties	Molecular Changes	Reference
hyperoxia 75%, P1–P7	mice	human	BMMSC; UCTMSC	exosomes	50 µL concentrate, equivalent to product of 5 × 10^5^ MSCs over 36 h	i.v.	P4	amelioration of alveolar simplification, fibrosis, pulmonary vascular remodeling by UCTMSC and BMMSC derived exosomes equally effective	amelioration of lung function and pulmonary hypertension	suppression of M1 macrophages, IL6, TNF-α, CCL2, CCL5, CCL7, augmentation of M2-like macrophages, CCL17	[97]
hyperoxia >95%, P1–P4	mice	human	UCTMSC	conditioned medium or exosomes	100 µL concentrate, equivalent to 7.6 × 10^5^ MSCs	i.p.	P2 and P4	improvements in lung and cardiac pathology	improved cardiovascular function	reduced influx of inflammatory cells, neutrophils	[99]
hyperoxia 75%, P1–P14	mice	human	UCTMSC	exosomes	100 µL concentrate, equivalent to 1 × 10^6^ MSCs	i.v.	P4 or 4 times on P18, P25, P32 and P39	in all application settings improvements in alveolar and vascular development, less fibrosis	improved exercise capacity, ameliorated pulmonary hypertension		[100]
hyperoxia 75%, P1–P14	mice	mice	BMMSC	conditioned medium	50 µL concentrate, equivalent to 5 × 10^4^ MSCs	i.v.	P4	preserved alveolar and vascular structures		reduced macrophages, neutrophils	[29]
hyperoxia 75%, P1–P14	mice	mice	BMMSC	conditioned medium	50 µL concentrate, equivalent to 5 × 10^4^ MSCs	i.v.	P4	preserved lung structure		increase in bronchioloalveolar stem cells	[68]
hyperoxia 75%, P1–P14	mice	mice	BMMSC	conditioned medium	50 µL (10µg MSC-CM protein), equivalent to 5 × 10^5^ MSCs	i.v.	P14	partially reversed alveolar and vascular injury, reversal of right ventricular hypertrophy	improved lung function, reversal of pulmonary hypertension, peripheral pulmonary vascular remodeling		[60]
hyperoxia 80%, P1–P14	rat	human	AMMSC	conditioned medium or exosomes	50 µL conditioned media or 300 ng exosomes in 50 µL, cell equivalent not available	i.t.	P7	amelioration of lung injury not as efficient as for AMMSC			[64]
hyperoxia 90%, P1–P14	rat	human	UCBMSC	exosomes	20 µg protein, derived from 5 × 10^5^ MSCs	i.t.	P5				[42]
hyperoxia 85%, P1–P14	rat	human	UCTMSC	exosomes	20 µg protein	i.t.	P7	restoration of alveolar structure	improved lung function	improved number of ki67, SPC and CD31 positive cells and VEGFA expression, reduced TUNEL positive cells, PTEN and cleaved caspase-3 expression	[101]
hyperoxia 60%, P1–P14	rat	human	UCTMSC	exosomes	50 µL concentrate, equivalent to 8 × 10^8^, 4.5 × 10^8^ and 3 × 10^8^ exosomes at P3, P7 and P10, cell equivalent not available	i.t.	P3, P7 and P10	better preserved alveolarization, vasculature development compared to MSC			[45]
hyperoxia 95%, P1–P15	rat	rat	BMMSC	conditioned medium	1 µL/g body weight concentrate, equivalent to 1.5 × 10^6^ MSCs	i.p.	daily P1–P21	better preserved alveolar growth, prevention of pulmonary hypertension			[96]
hyperoxia 90%, P2–P16	rat	rat	BMMSC	conditioned medium	50 µL concentrate, equivalent to 2 × 10^6^ MSCs	i.t.	P9	improvements in alveolar and vascular development, conditioned medium as effective as MSC		reduced IL1β,IL6, preserved Ang-1, VEGFA	[53]
hyperoxia 85%, P1–P14	rat	rat	BMMSC	exosomes	50 µL concentrate (3.4 × 10^9^ exosomes, product of approx. 10^6^ cells)	i.p.	daily P2-P15	protection of alveolarization, angiogenesis and reduction of right heart hypertrophy			[98]
hyperoxia 95%, P1–P14	rat	rat	UCBMSC	conditioned medium	7 µL/g body weight; harvested at 90% cell confluency, cell equivalent not available	i.t.	daily P4-P21 or daily P14-P28	similarly attenuated alveolar and vascular changes as by early and late MSC	similarly improved exercise capacity		[61]

MSC—mesenchymal stem cells; BPD—bronchopulmonary dysplasia; Pn—postnatal day n; AMMSC—amniotic membrane mesenchymal stem cells; BMMSC—bone marrow mesenchymal stem cells; UCBMSC—umbilical cord blood mesenchymal stem cells; UCTMSC—umbilical cord tissue mesenchymal stem cells; i.v. —intravenous; i.t. —intratracheal; i.p.—intraperitoneal; CM—conditioned medium; IL—interleukin; TNF—tumor necrosis factor; VEGFA—vascular endothelial growth factor; Ang—angiopoietin; CCL—chemokine (C-C motif); SPC—surfactant protein C; TUNEL—TdT-mediated dUTP-biotin nick end labeling; PTEN—phosphatase and tensin homolog.

**Table 6 ijms-22-01138-t006:** Evaluation of additional benefits of MSC application in combination with further treatment approaches in the BPD rodent model.

Experimental Lung Disease Model	Species	Cell Source	MSC Species	Modification	Dose (Cells)	Application Route	Time Point of Application	Histologic Evaluation	Functional Properties	Molecular Changes	Reference
hyperoxia 60%, P1–P14	mice	mice	BMMSC	MSC plus recombinant erythropoetin	1 × 10^6^ MSCs and 5000 U/kg EPO	i.v.	P1 and P7	improved airway structures		improved PECAM, VEGFA, reduced MMP-9	[36]
hyperoxia 85%, P1–P14	rat	human	PTMSC	MSC plus natural surfactant	1 × 10^5^ MSCs and 10 µL surfactant (corr. to 35 mg/kg phospholipids)	i.t.	P5	no additional benefit for alveolar hypoplasia			[39]

MSC—mesenchymal stem cells; BPD—bronchopulmonary dysplasia; Pn—postnatal day n; BMMSC—bone marrow mesenchymal stem cells; PTMSC—placental tissue mesenchymal stem cells; i.v.—intravenous; i.t.—intratracheal; EPO—erythropoietin; MMP—matrix metalloproteinase; PECAM—platelet endothelial cell adhesion molecule; VEGFA—vascular endothelial growth factor.

**Table 7 ijms-22-01138-t007:** Impact of MSC preconditioning on efficacy in the BPD rodent model.

Experimental Lung Disease Model	Species	Cell Source	MSC Species	MSC Preparation	Modification	Dose (Cells)	Application Route	Time Point of Application	Histologic Evaluation	Functional Properties	Molecular Changes	Reference
hyperoxia 95%, P1–P15	rat	rat	BMMSC	conditioned medium	hyperoxic preconditioning of MSC	1 µL/g body weight equivalent to 1.5 × 10^6^ MSCs	i.p.	daily P1–P21	better preserved alveolar growth, prevention of pulmonary hypertension			[96]

MSC—mesenchymal stem cells; BPD—bronchopulmonary dysplasia; Pn—postnatal day n; BMMSC—bone marrow mesenchymal stem cells; i.p.—intraperitoneal.

**Table 8 ijms-22-01138-t008:** Effects of MSC transduction before application in the BPD rodent model.

Experimental Lung Disease Model	Species	Cell Source	MSC Species	MSC Preparation	Modification	Dose (Cells)	Application Route	Time Point of Application	Histologic Evaluation	Functional Properties	Molecular Changes	Reference
hyperoxia >95%, P1–P4	mice	human	UCTMSC	exosomes	MSC with knockdown of TSG-6	100 µL concentrate, equivalent to 7.6 × 10^5^ MSCs	i.p.	P2 and P4	abrogation of improvements in lung and cardiac pathology		influx of inflammatory cells and neutrophils no longer inhibited	[99]
hyperoxia 90%, P1–P14	rat	human	UCBMSC		MSC with knockdown of decorin	1 × 10^5^	i.t.	P5	attenuated beneficial effects on alveolar structures		reverted IL6, IL8 upregulation, reduced IL10, M1 macrophage polarization retained	[46]
hyperoxia 90%, P1–P14	rat	human	UCBMSC		MSC with knockdown of PTX3	1 × 10^5^	i.t.	P5	attenuation of reduced lung injury		attenuation of macrophage shift from M1 to M2, IL10, preserved IL6, IL8	[49]
hyperoxia 90%, P1–P14	rat	human	UCBMSC		MSC with knockdown of VEGFA	5 × 10^5^	i.t.	P5	beneficial effects on alveologenesis, vasculogenesis abolished		reduced apoptosis, macrophages, IL1α, IL1β, IL6, TNFα abolished	[56]
hyperoxia 90%, P1–P14	rat	human	UCBMSC	exosomes	MSCs with knockdown of VEGFA	20 µg EVs, derived from 5 × 10^5^ MSCs	i.t.	P5	attenuation of beneficial effects on alveolarization, angiogenesis		attenuation of beneficial effects on cell death, activated macrophages, s IL1α, IL1β, IL6, TNF-α	[42]
hyperoxia 80%, P1–P15	rat	rat	BMMSC		MSC transfected with 7ND-CCL2	1 × 10^5^	i.v.	P5	improved alveolar and vascular structures	reduced pulmonary hypertension	reduced M1 macrophages, IL6	[76]
hyperoxia 85%, P1–P21	rat	rat	BMMSC		MSC with knockdown of SDF-1	1 × 10^6^	i.t.	P7	attenuation of all beneficial MSC effects		reduced inflammatory macrophages, IL1β and improved IL10 prohibited	[38]

MSC—mesenchymal stem cells; BPD—bronchopulmonary dysplasia; Pn—postnatal day n; BMMSC—bone marrow mesenchymal stem cells; UCBMSC—umbilical cord blood mesenchymal stem cells; UCTMSC—umbilical cord tissue mesenchymal stem cells; ATMSC—adipose tissue mesenchymal stem cells; i.v. —intravenous; i.t. —intratracheal; i.p. —intraperitoneal; EV—extracellular vesicles; IL—interleukin; TNF—tumor necrosis factor; TSG—TNF-stimulated gene; VEGFA—vascular endothelial growth factor; CCL—chemokine (C-C motif); PTX—pentraxin; SDF—stromal cell-derived factor.

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
