# Peer review of "MSC Based Therapies to Prevent or Treat BPD—A Narrative Review on Advances and Ongoing Challenges"

_ijms, 2021, doi:10.3390/ijms22031138_

Round 1

Reviewer 1 Report

In the manuscript entitled : „MSC based therapies to prevent or treat BPD- a narrative review on advances and ongoing challenges”, the Authors analyze numerous studies (both preclinical, and early phase clinical) on therapeutic potential of mesenchymal stem cells as an option of treatment in children with bronchopulmonary dysplasia. The Authors have done a great job – their overwiev covers probably all studies in this area, that have been published during the last few years. Manuscript is well organized, easy to read despite of its lenght. The rationality of the review is very good presented.

Author Response

We highly appreciate the reviewers’ positive feedback to our review.

Reviewer 2 Report

The authors developed an actuallized and comprehensive review about the state of art on mesenchymal stem-cell therapy in bronchopulmonary dysplasia.
In my opinion the paper is adequate to be published in the present form.

Author Response

We highly appreciate the reviewer's positive feedback to our review.

Reviewer 3 Report

This is a well-documented review on the use of MSCs for the experimental treatment of bronchopulmonary dysplasia, a disease for which no definite cure has been found so far. This review is of current interest, as shown by several ongoing clinical trials on this topic. However, some statements could be misleading and should be reassessed.

Lines 191-192. Please report the dose per Kg body wt, since cited experiments include both mice and rats.

Lines 384-385. This is a strong statement: to my knowledge there is no experimental evidence in vivo that MSCs can aggravate fibrosis. The cited reference is a review; please refer to the original paper, if any.

Lines 403-404. There is no evidence that EVs contain “all” active factors.

Lines 430-431. The sentence is misleading: the cited reference reports biodistribution following intratracheal (not intravenous) administration. EV uptake by endothelial cells has been well documented in different experimental settings.

Lines 447-450. This sentence is too simplistic and misleading. The Authors should consider that the doses administered to experimental animals are ten to hundred times higher than those administered to human patients.

Lines 450-451. This sentence is not supported by any available data. On the contrary, according to some companies active in the field, the cost of producing EVs can be significantly lower when compared to MSCs: see e.g. Nature Biotechnology 37 (2019) 1395.

Lines 475-477. The Authors are comparing several studies performed in animals with only two trials performed in preterm infants, where the heterogeneity of results could be more likely ascribed to differences in disease severity, as reported previously.

Lines 535-536. Better refer to [“Complex interactions between” inflammatory injury and disruption …] to justify the use of [broad-acting therapeutics].

Lines542-543. This sentence is unclear.

Lines 553-554. Please explain what you mean with “standardization of experimental approaches”.

Lines 555-557. How do you justify the complexity and especially the cost of the primate model? Do you really think that such an approach could be feasible?

Author Response

We highly appreciate the reviewer's positive estimate and thank for the valuable comments that we integrated into the revised version.

Lines 191-192. Please report the dose per Kg body wt, since cited experiments include both mice and rats.

Response and changes to the manuscript:

As suggested by the reviewer, we detailed the doses applied per animal for both citations.

Line 192-195. ” …at a dosage of 3 to 6 x105 cells given intratracheally per animal. Of notice, improvements in lung structures persisted into adulthood and the systematic review confirmed efficacy in mice and rats within a dose range from 5x104 to 5x106 cells per animal.”

Lines 384-385. This is a strong statement: to my knowledge there is no experimental evidence in vivo that MSCs can aggravate fibrosis. The cited reference is a review; please refer to the original paper, if any.

Response and changes to the manuscript:

We thank the reviewer for his comment and deleted the sentence in line 384-385.

Lines 403-404. There is no evidence that EVs contain “all” active factors.

Response and changes to the manuscript:

Again we thank the careful review of our manuscript and substituted all by many

Line 403. “many factors of MSC”

Lines 430-431. The sentence is misleading: the cited reference reports biodistribution following intratracheal (not intravenous) administration. EV uptake by endothelial cells has been well documented in different experimental settings.

Response and changes to the manuscript:

We revised the statement as suggested by the reviewer.

Line 430-431. “The incorporation of EV is not cell type specific and EV were verified in type II cells, in lung fibroblasts and pericytes when given intratracheally [42].”

Lines 447-450. This sentence is too simplistic and misleading. The Authors should consider that the doses administered to experimental animals are ten to hundred times higher than those administered to human patients.

Lines 450-451. This sentence is not supported by any available data. On the contrary, according to some companies active in the field, the cost of producing EVs can be significantly lower when compared to MSCs: see e.g. Nature Biotechnology 37 (2019) 1395.

Response and changes to the manuscript:

We thank the reviewer for his feedback and rephrased the sentence as suggested.

Line 447-450: “The former obstacles of having high amounts of EV available for the conduction of clinical studies in the preterm infant, are in the meantime overcome by the rapid increase in companies stepping into the field of EV production and provision for clinical trials [122].”

Lines 475-477. The Authors are comparing several studies performed in animals with only two trials performed in preterm infants, where the heterogeneity of results could be more likely ascribed to differences in disease severity, as reported previously.

Response and changes to the manuscript:

As suggested by the reviewer, we added further important reasons for the lack of evidence in the two clinical trials.

Line 476-477. “…., the heterogeneity of BPD severity in participating patients and the approach of clinical studies to demonstrate safety but not efficacy,…”

Lines 535-536. Better refer to [“Complex interactions between” inflammatory injury and disruption …] to justify the use of [broad-acting therapeutics].

Response and changes to the manuscript:

As suggested by the reviewer, we rephrased the sentence.

Line 536. “Complex interactions between…”

Lines 542-543. This sentence is unclear.

Response and changes to the manuscript:

We thank the reviewer for the hint and specified the sentence.

Line 544-548. “As for all other new therapies, a careful and complete evaluation of MSC-based approaches covering i.e. the lung microenvironment and the comprehensive documentation of side effects under particular conditions like the recently described increased pulmonary artery embolism in a sheep ECMO model of ARDS are indispensable [118,119].”.

Lines 553-554. Please explain what you mean with “standardization of experimental approaches”.

Response and changes to the manuscript:

We included details on our statement.

Line 559-560. “…with respect to dosing, timing and selection of precisely defined patient groups…”

Lines 555-557. How do you justify the complexity and especially the cost of the primate model? Do you really think that such an approach could be feasible?

Response and changes to the manuscript:

We now refer to the ongoing research project in the preterm baboon model that has been published so far only as an abstract.

Line 560-561. “Ongoing investigation within the baboon model of borderline viability are suitable to confirm….”